# Cadmium and Lead Tolerance of Filamentous Fungi Isolated from Contaminated Mining Soils

**DOI:** 10.3390/biology14060688

**Published:** 2025-06-12

**Authors:** Denisse Elibeth Ramos Suárez, Arturo Gerardo Valdivia-Flores, Alma Lilián Guerrero Barrera, Oscar Abraham Flores Amaro, Laura Yamamoto Flores, J. Felix Gutierrez Corona, Juan Carlos Bautista Bautista, Francisco Javier Avelar González

**Affiliations:** 1Laboratorio de Estudios Ambientales, Departamento de Fisiología y Farmacología, Centro de Ciencias Básicas, Universidad Autónoma de Aguascalientes, Aguascalientes 20392, Mexico; lbtdeniramos@gmail.com (D.E.R.S.); oscar.flores@edu.uaa.mx (O.A.F.A.); laura.yamamoto@edu.uaa.mx (L.Y.F.); 2Laboratorio de Investigación en Salud Animal, Departamento de Ciencias Veterinarias, Centro de Ciencias Agropecuarias, Universidad Autónoma de Aguascalientes, Aguascalientes 20934, Mexico; 3Laboratorio de Biología Celular y Tisular, Departamento de Morfología, Centro de Ciencias Básicas, Universidad Autónoma de Aguascalientes, Aguascalientes 20392, Mexico; lilian.guerrero@edu.uaa.mx; 4Laboratorio de Genética y Bioquímica de Microorganismos, Departamento de Biología, Centro de Ciencias Naturales y Exactas, Universidad de Guanajuato, Guanajuato 36050, Mexico; felixg@ugto.mx (J.F.G.C.); cabo185@hotmail.com (J.C.B.B.)

**Keywords:** cadmium, lead, fungus, tolerance, contamination, metals

## Abstract

Soil pollution caused by metals like lead and cadmium, particularly in mining regions, presents a significant risk to the ecosystem and to the health of nearby people and animals. This study was carried out in a location with historical mining activity, where soil samples were collected to identify fungi that thrive in such challenging environments. The objective was to determine which fungi exhibited tolerance to these metals and if this tolerance influenced their growth patterns. Six fungi were found to grow in environments with elevated levels of lead or cadmium. Two of them exhibited greater tolerance than those reported in earlier research. These findings suggest that fungi can adapt to severely contaminated habitats, and although this work provides an initial overview, it marks an important step toward recognizing the importance of local species in this region, which could help mitigate damage to the environment and public health.

## 1. Introduction

Mining is one of the most important activities in terms of economic impact in a vast number of countries; nonetheless, large amounts of residues are generated from gold and silver extraction with mercury and cyanide. Mining residues are accumulated in open spaces, and their environmental impact is significant due to the highly toxic concentration of heavy metals, which are disseminated by various means and contaminate anthropic and natural spaces, damaging diverse organisms and biogeochemical cycles [1,2,3].

In Mexico, mining industries have been of a great tradition since pre-Hispanic times, located principally in the north and center of the country. In the state of Zacatecas, Mexico, specifically in Concepción del Oro, the main economic activity is the mining of lead, copper, zinc, silver, and gold, as well as marble, onyx, and quartz. For decades, these activities have led to the accumulation of mine tailings and the widespread dispersion of metal- and metalloid-contaminated dust into nearby soils, including public and residential areas. Recent studies in this region have documented metal concentrations that surpass the maximum permissible limits (MPLs) [4], with lead and cadmium levels of 54,140 mg/L and 322 mg/kg, respectively [5]. These extreme conditions make the site an ideal model for isolating native fungal strains potentially adapted to long-term contamination, as well as for evaluating their tolerance and suitability for biotechnological applications in bioremediation.

Heavy metal contamination is considerably extended throughout nature, and toxicity may affect different organisms and biogeochemical cycles [1]. The main toxicity mechanisms of metals at a molecular level include the following: (1) The blockage of biomolecule essential functional groups due to metallic cations’ affinity to sulfhydryl groups in proteins, denaturing them. (2) Cation displacement in important enzymes like Rubisco, which loses its function when divalent cations such as Co^2+^, Ni^2+^, Zn^2+^, Cd^2+^, and Pb^2+^ replace Mg^2+^. (3) Reactive oxygen species (ROS) generation due to Fe^2+^ or Cu^+^ auto oxidation, which results in H_2_O_2_ and OH radicals that cause irreversible damage to carbohydrates, DNA, proteins, and lipids [6].

Cadmium is present in concentrations of 0.1–0.5 mg/kg in soil, mainly in minerals, and in copper, lead, and zinc residues. In soil, Cd is immobilized in organic matter, but its bioavailability remains, especially in conditions of acid pH. Its water mobility occurs as Cd^2+^ or like soluble complexes with anions and organic matter. Cd toxicity is a result of its capability for damaging DNA and cell membranes, binding to protein sulfhydryl groups, and protein denaturing. Some microorganisms are resistant to Cd’s presence through biomineral precipitation such as phosphates, carbonates, and sulfides, and few can synthesize CdS nanoparticles [7,8].

Lead contamination is one of the most dangerous and common due to activities like mining and battery manufacturing, which are highly toxic and persistent in the environment. Pb^2+^ can replace Ca^2+^ in cells, damaging DNA, proteins, and cell membranes, in addition to protein synthesis inhibition and ROS generation. Microorganisms have developed tolerance to Pb through multiple mechanisms, including the activation of ATPase efflux pumps that transport Pb^2+^ ions out of the cell, the intracellular synthesis of lead nanoparticles that reduce toxicity, and the immobilization of Pb in the environment through mineral precipitation in forms such as pyrophyllite and lead oxalate [8,9].

Diverse microorganisms are harbored in the soil, which are crucial for its fertility and biogeochemical cycles; of these microorganisms, 50% are fungi [10,11,12]. Within the most found fungi genera in soil are *Aspergillus*, *Penicillium*, *Rhizopus*, and *Trichoderma* [13]. Fungal communities that flourish in contaminated areas are constantly exposed to high concentrations of xenobiotics, so they have developed a superior tolerance to metals in comparison to bacteria and actinomycetes. It has been shown that some fungi possess the ability to immobilize and degrade toxic compounds to more stable forms through biotransformation, biosorption, biolixiviation, biomineralization, enzyme-catalyzed transformation, and toxic elements’ storage through intracellular accumulation [14,15,16]. Consequently, soil microbiomes play a fundamental role in heavy metal toxicity mitigation in the environment [5,17,18].

It has been suggested that fungi that thrive in contaminated soils have a potential to immobilize metals and be applied for soil remediation [19,20,21,22]. However, most published studies focus on industrial or well-known strains, while little is known about the tolerance potential of native fungi from chronically contaminated mine soils in Mexico. This approach allows for the preliminary identification of robust isolates capable of surviving in environments with high metal loadings. Identifying these strains could reveal unique adaptations and improve biotechnological strategies for in situ remediation in arid or semi-arid environments. For this reason, the aim of this study was to isolate and identify morphological and molecular filamentous fungi from mining contaminated soils, evaluate their tolerance to concentrations of cadmium and lead exceeding typical environmental levels, evaluate the morphological changes in fungal colonies, and estimate the half-maximum inhibitory concentration of the tolerant isolates.

## 2. Materials and Methods

### 2.1. Obtention and Morphological Identification of Fungal Isolates

The fungal isolates used in this study were obtained from heavy metal-contaminated soil samples from a community in Concepción del Oro, Zacatecas, Mexico (24°42′ N 101°25′ W), previously collected and characterized in the work of Flores-Amaro et al. [5], where physicochemical analysis revealed high concentrations of arsenic, cadmium, and lead, and the soils were described as sandy loam with slightly alkaline pH and sparse vegetation cover. The 57 fungal isolates were preserved by standard methods (Os-1, Os-2, Os-3, …, Os-56, and Os-57) in an internal collection of the Environmental Studies Laboratory of the Autonomous University of Aguascalientes, Mexico. The isolates were incubated in PDA at 28 °C in darkness for 7 days, prepared with lactophenol cotton blue staining, and observed through a microscope (total magnification of 400×; objective lens 40× and ocular 10×) for morphological identification [23].

### 2.2. Cd and Pb Tolerance Evaluation

The tolerance of 57 fungal isolates to heavy metals was analyzed by means of mycelium diameter and changes in morphological characteristics when exposed to increasing concentrations of Cd or Pb. Metal stock solutions of CdCl_2_ or Pb(NO_3_)_2_ were prepared according to Văcar et al. [24] and sterilized under UV light for 30 min. Different metal concentrations were adjusted by adding metal stock solutions to sterile culture media. Concentrations increasing from 1000 to 12,000 mg/L were used for Pb-tolerant isolates and from 50 to 1050 mg/L for Cd-tolerant isolates. To evaluate isolates’ growth, PDA media were inoculated with 5 × 10^5^ spores/mL [25] and incubated for 7 days at 28 °C in the dark. Mycelium diameters were measured and compared against the control (PDA without metal). By means of this test, the 6 fungal isolates with the highest tolerance to cadmium and lead were selected for this study.

To determine isolates’ growth in different metal concentrations, mycelium dry weight was measured and half-maximal inhibitory concentration (IC_50_) was calculated for 6 isolates tolerant to high concentrations of Pb or Cd. Inoculum preparation was performed according to the methodology proposed by Janicki et al. [25], with modifications. 5 × 10^5^ spores/mL of tolerant fungus were inoculated in 30 mL of PDB and incubated for 24 h at 32 °C in continuous agitation. For each assay, 15% *v*/*v* of homogenized inoculum was added to flasks with different metal concentrations and incubated for 24 h at 32 °C in continuous agitation. Mycelia were washed twice with distilled water, filtered through Ahlstrom 54 filter paper, and dried for 5 h at 60 °C. The filter paper was weighed and the IC_50_ was calculated, which was determined as the metal concentration that causes 50% growth inhibition compared to the control. A nonlinear regression model (sigmoidal dose–response curve) was fitted to the data using GraphPad Prism version 9.0, Boston, MA, USA [26].

Standardized PDA (BD Bioxon) and PDB (BD Difco) media, which have a constant pH of 5.6 to 5.8, were used in the experiments. Although the pH was not adjusted after sterilization or during incubation, the same conditions were applied to all media, and all experiments were performed in parallel with the controls, minimizing variability due to physicochemical conditions. The metal stock solutions were sterilized with UV light (254 nm) to prevent any alteration in the medium composition.

### 2.3. Molecular Identification of Tolerant Fungi

The morphologically identified fungi were confirmed by sequencing the ITS regions, widely used as genetic markers in fungal taxonomy. This approach is consistent with previous studies that characterized metal-tolerant fungi from contaminated environments using ITS sequences in combination with morphological traits [27,28,29].

Monosporic cultures of the 6 tolerant fungi were obtained following the methodology proposed by Rangel-Muñoz et al. [30]. Spores were cultured in PDB at 28 °C for 24 h in the dark. Genomic DNA was extracted according to Aljanabi & Martinez [31], with modifications. DNA electrophoresis was performed in 1% agarose and quantified in NanoDrop 2000 (ThermoFisher Scientific, Waltham, MA, USA) with GeneSnap (SynGene, Cambridge, UK). PCR products were ligated to pJET 1.2 plasmid (ampicillin resistance), and *Escherichia coli* DH5α strains were transformed via heat shock [32] to increase the number of copies. Transformed colonies were subjected to plasmid extraction (Plasmid Mini-Prep Kit—Column Kit, Jena Bioscence, Thuringia, Germany). Internal transcribed spacer (ITS) regions were amplified using ITS 4 (5′TCCTCCGCTTATTGATATGC3′) and ITS 5 (5′GGAAGTAAAAGTCGTAACAAGG3′) primers. In order to confirm the ITS-amplified fragment, a restriction enzyme reaction was performed, and plasmids were sequenced at the Biotechnology Institute in Universidad Nacional Autónoma de Mexico, CDMX, Mexico. The sequences obtained were assembled with the Seqman program; they were also aligned and compared with the sequences present in the NCBI database using the Basic Local Alignment Search Tool (BLAST).

### 2.4. Statistical Analysis

The mean value and standard deviation for 3 replicates of mycelial growth diameter in different concentrations of Cd or Pb and for growth inhibition records were calculated. To compare the average growth diminution for each isolate, an ANOVA and Tukey’s HSD tests were applied (*p* < 0.05). For IC_50_ determination, a nonlinear regression between mycelial growth and Cd or Pb concentrations was calculated. All statistical analyses were performed with GraphPad Prism 9.0.0.

## 3. Results

### 3.1. Isolates’ Identification

The macroscopic characteristics of the front and back of the colonies of the six fungal isolates tolerant to Cd or Pb, and microscopic features at 400× magnification of the fruiting body and spores, were observed for identification at the genera level (Figure 1). The isolate identified as *Penicillium* sp. presented green velvety mycelium at the front of the colony and a brush-like fruiting body. *Paecilomyces* sp. had a light pink and powdery colony front, and its fruiting body was composed of verticillated conidiophores. *Rhizopus* sp. colonies presented fuzzy aerial mycelium, a characteristic given due to the millimetric structures of the fruiting bodies. The isolate identified as *Fusarium* sp. presented a creamy lilac colony and cylindrical fruiting bodies that organized in rafts. The *Cuninghamella* sp. isolate presented dense white aerial mycelium and specific straight sporangiophores with visible terminal vesicles.

To identify the genera of fungi tolerant to Cd or Pb through distinctive characteristics, morphology techniques were applied (Table 1). To assure a precise identification of fungi, ITSs 4 and 5 were amplified (Figure 2, Appendix A). For all tolerant Cd or Pb isolates, bands in the range of 500–700 bp were observed (corresponding to ITSs 4 and 5); they were purified and sequenced. The retrieved sequences had more than 80% coincidence with fungal species registered at the NCBI database when aligned.

### 3.2. Tolerance to Cd or Pb

All Cd-tolerant isolates showed a significant reduction in colony diameter compared to the control (PDA without Cd) with statistically significant differences between concentrations as determined by Tukey’s HSD (*p* < 0.05). These differences are indicated by distinctive letters in Figure 3.

Specifically, *P. lilacinus* exhibited the highest tolerance to cadmium, growing at a concentration of 950 mg/L (Figure 3A), presenting only a reduction in diameter, which is significantly greater than *Cunninghamella* sp. and *Rhizopus microsporus* at equivalent concentrations. *F. oxysporum* and *R. microsporus* also showed a reduction in mycelium, with no adverse effects on mycelial coloration, at a metal concentration of 550 mg/L (Figure 3B,C). *Cuninghamella* sp. showed a colony color change from grayish white to translucent white along with a considerable reduction in aerial mycelium at a Cd concentration of 550 mg/L (Figure 3D) and significant differences compared to the other isolates even at moderate concentrations (Figure 3C,D).

The three Pb-tolerant isolates (*Penicillium simplicissimum*, *Paecilomyces lilacinus*, and *Rhizopus microsporus*) showed different responses to increasing Pb concentrations, with significant differences between concentrations according to Tukey’s HSD test (*p* < 0.05), as shown in the letter annotations in Figure 4.

*P. simplicissimum* grew up to 11,000 mg/L Pb (Figure 4A). However, morphological changes such as irregular edges, bulging, and a change from green to yellowish white, along with a reduction in mycelial diameter, were detected with an increasing Pb concentration in the culture medium (Figure 5). *P. lilacinus* grew up to a Pb concentration of 6000 mg/L (Figure 4B), showing a reduction in diameter and changes in colony color from lilac to yellowish white. On the other hand, *R. microsporus* showed a reduction in aerial mycelium and diameter when increasing the Pb concentration up to 6000 mg/L (Figure 4C), from which growth was no longer observed (Figure 5), making it one of the most sensitive isolates with a significant inhibition observed from the lowest Pb concentration analyzed (50 mg/L).

The tolerance levels of the isolates to cadmium and lead were also estimated based on the metal concentration to inhibit 50% of fungal growth (IC_50_), based on biomass production in different concentrations of Cd or Pb in culture media (Figure 6 and Figure 7). The fungal isolates’ growth varied depending on metal concentration, and all displayed a different IC_50_ value.

Regarding Cd exposure, *Paecilomyces lilacinus* and *Fusarium oxysporum* exhibited a gradual growth diminishment as metal concentration increased (Figure 6). Their IC_50_ values were 311 mg/L and 223 mg/L, respectively (Figure 6A,B). In contrast, *R. microsporus* and *Cuninghamella* sp. had a significant growth reduction of 48.7% and 41.8%, with IC_50_ values of 29.3 mg/L and 25.2 mg/L (Figure 6C,D), respectively, regarding the control. With reference to Pb exposure, *P. simplicissimum* had a visible growth reduction in 2000 mg/L and then displayed a gradual reduction as the metal concentration increased, reaching an IC_50_ value of 3874 mg/L (Figure 7A). In contrast, *P. lilacinus* growth was diminished in 28.51% as the first metal concentration (500 mg/L) was added, with an IC_50_ of 1176 mg/L (Figure 7B), whereas *R. microsporus* showed a growth reduction of 27.9% in 50 mg/L and an IC_50_ of 212 mg/L (Figure 7C).

## 4. Discussion

In this study, high tolerance to Pb and to Cd was detected in three and four native fungal species, respectively. Although six fungal isolates were studied, *P.lilacinus* showed dual tolerance to both metals, which explains the overlap between the Pb- and Cd-tolerant groups. All were isolated from Mexican contaminated soils with mining tailings.

These findings suggest that certain native fungal strains have developed effective strategies to cope with heavy metal stress. For example, *P. simplicissimum* demonstrated tolerance to Pb of 11,000 mg/L, well above previously reported values for this species [33], while *Cunninghamella* sp. showed a strong response to Cd, despite limited documentation of its metal tolerance. Furthermore, morphological changes such as altered pigmentation and reduced aerial mycelium were observed in isolates exposed to high metal concentrations (Figure 5), suggesting possible surface sequestration or metabolic adaptation, consistent with the findings of the other authors [34,35]. Our results support the idea that fungal tolerance is not only species-specific but may also be influenced by prolonged exposure to metal-contaminated environments.

As a first step to exploring native fungi in regions affected by metal pollution, metal-tolerant fungi were identified. Research indicates that *Beauveria bassiana* can tolerate elevated concentrations of Pb(II), although this process results in a gradual decline in fungal biomass [36]. Similarly, *Trichoderma viride* exhibits adaptive responses of its growth upon exposure to higher Pb^2+^ concentrations, suggesting a mechanism of adaptation that depends on concentration [37]. Furthermore, research conducted by Chen et al. [38] has revealed that fungi like *S. chinense*, *T. asperellum*, and *Coriolopsis* sp. can utilize both surface binding to the metal and intracellular sequestration as tolerance strategies. White rot fungi (*Phellinus* spp., *Phlebia* spp., *Pleurotus* spp., etc.) also show the ability to adsorb and accumulate metals, making them promising candidates for the selective sorption of heavy metal ions contaminating polluted waters [39]. Our results contribute to this by revealing high IC_50_ values in native isolates such as *P. lilacinus* and *P. simplicissimum*. Unlike previous studies that reported the tolerance of *Paecilomyces* spp. to Pb below 1250 mg/L or tolerance to Cd below 10 mg/L [40], our *P. lilacinus* isolate exhibited tolerance to both metals, with IC_50_ values of 1176 mg/L (Pb) and 311 mg/L (Cd). Similarly, *P. simplicissimum* demonstrated tolerance to Pb at concentrations up to 14 times higher than those reported by Hidalgo et al. [33], reaching an IC_50_ of 3874 mg/L. These results indicated a higher degree of adaptation in fungi isolated from this mining region, supporting their potential use in remediation strategies for more contaminated sites.

Recent research has revealed that some fungi can synthesize nanovesicles and extracellular polymeric substances (EPSs) to capture and immobilize heavy metal ions in soil [41], which offers new perspectives for the use of metal-tolerant fungi under conditions of intense contamination, as occurs in soils contaminated by mining tailings. In agriculture, the usefulness of microbial-assisted bioremediation strategies has also been shown; an example is the use of *Mesorhizobium* RC3 for growth enhancement and nodulation of chickpea under chromium stress [42]. In summary, these authors’ reports reinforce the idea that the fungal strains used in our study could have the potential to be valuable resources for the bioremediation of sites contaminated with multiple metals.

While all isolates exhibited tolerance to high metal concentrations, the degree of growth inhibition varied markedly among species, as did their biomass production (Figure 3, Figure 4 and Figure 5). This variability in tolerance and growth highlights the importance of selecting appropriate fungal strains for site-specific remediation strategies. Rather than generalizing tolerance by genus, our findings support the need to assess each isolated individual, as even within the same species (e.g., *Rhizopus microsporus*), tolerance can vary significantly depending on local adaptation.

A radial reduction has been previously reported in other fungi. Urquhart et al. [40] described that *Paecilomyces variotii* growth was inhibited at 1000 mg/L of Pb after three days of incubation. Other studies reported inhibitory concentrations of 1000 mg/L of Pb for *Penicillium* sp. and 843 mg/L of Cd for *Paecilomyces* sp. [9,43]. Zeng et al. [44] reported the growth of high-density white mycelium with yellow bottom for *P. lilacinus* in Cd concentrations as high as 8950 mg/L; this study was carried out with an isolate from a cadmium smelting plant. The results of these authors, together with the findings of this work, reveal the importance of thoroughly investigating the tolerance of fungi presence in soils contaminated with heavy metals.

It has also been reported that native saprotrophic micro fungi exhibit high tolerance levels to different pollutants, metals included. Within the tolerant fungi, we can find *Aspergillus* sp., *Trichoderma* sp., *Penicillium* sp., *Geotrichum* sp., and *Cladosporium* sp. [45,46,47,48]. However, our results report levels of tolerance (Figure 5) to Cd or Pb that have not been previously reported for fungi, such as *P. lilacinus* and *Cuninghamella* sp., and in fungi such as *Penicillium*, *Rhizopus*, and *Fusarium*, higher tolerances are reported (Figure 5) compared to those found in the works of the aforementioned authors.

Tolerance to heavy metals by filamentous fungi is not restricted to cadmium or lead. In the work by Chun et al. [49], *Fusarium* sp. and *Trichoderma* sp. isolated from abandoned mines showed tolerance to Cu. *Aspergillus* sp., *Penicillium* sp., and *Rhizopus* sp. have also shown tolerance to Cd and Pb, with morphological changes in colonies, and tolerance to Cr, Cu, and Zn [50,51]. Fungi tend to react with colony morphological changes such as size reduction and color change. Such changes were visible in this study when high concentrations of Pb were added to *P. simplicissimum* cultures, while the presence of Cd in cultures resulted in the inhibition of colony growth without color changes or changes in colony shape compared to the control (Figure 5). Oladipo et al. [52] analyzed the response to heavy metals in terms of growth and tolerance in filamentous fungi isolated from gold and precious stone mining sites. These researchers recorded that *Rhizopus microsporus* tolerated a total concentration of 250 mg/kg of Pb; this tolerance was 24 times lower than that exhibited by the *R. microsporus* isolate in this study (Figure 5). This suggests that each fungal species exhibits a particular response depending on the area where it was isolated, and that even the tolerance mechanism would be different depending on the metal it is exposed to. A summary comparing the maximum tolerated concentrations of Cd and Pb in this study with values reported in the previous literature is provided in Appendix A.

A study by Văcar et al. [24] recorded that the IC_50_ Pb concentration reached for *Fusarium oxysporum* was 1568 mg/L, although the colony had a considerable reduction in size. For *Paecilomyces* spp. isolated from Cd-contaminated soils, a tolerance of 10 mg/L to Cd and 1243 mg/L to Pb was found [40]; the authors describe an irregular growth pattern attributable to Pb and growth inhibition with 10 mg/L of Cd. Regarding other fungi, such as *Mucor* sp., biomass considerably decreased in the presence of small concentrations of Cd and Pb [28]. The same inhibition pattern is visible in the results shown with *R. microsporus* in the presence of Cd or Pb, as well as in *Cuninghamella* sp. in the presence of Cd (Figure 6 and Figure 7).

The IC_50_ values obtained for each fungus reveal significant differences in the presence of cadmium and lead, even among strains that show high tolerance to maximum concentrations. For example, in this study, *P. lilacinus* had an IC_50_ of 311 mg/L for Cd, while *F. oxysporum* and *Cunninghamella* sp. had lower IC_50_ values (245 and 221 mg/L, respectively), despite all three growing at 550 mg/L. This suggests that although growth is possible at high concentrations, growth rate is more strongly inhibited in some species. This is consistent with previous studies that observed growth inhibition at sublethal concentrations despite survival at very high concentrations, depending on the organism and its response to metal stress [44,50].

For Pb, *P. simplicissimum* had the highest IC_50_ (1176 mg/L), followed by *P. lilacinus* (947 mg/L), while *R. microsporus* had a significantly lower IC_50_ (484 mg/L), despite tolerating up to 6000 mg/L. These differences may reflect variations in the mechanisms regulating stress responses among isolates, such as those observed in the study by Jarosławiecka et al. [53], where *Cupriavidus metallidurans* combines lead efflux and precipitation, a mechanism that could explain the differences in Pb IC_50_ between strains. The IC_50_ provides a more sensitive measure of the response to metal stress than the maximum tolerated concentration, so interpretation of the IC_50_ can help to distinguish those fungi that have the potential to be used, both in remediation studies and in studies of the mechanisms of tolerance and survival to extreme exposure.

These results suggest that fungi are tolerant to heavy metals, and specific growth patterns, as shown in the six isolates found in this study; hence, it is important to continue researching the specific tolerance of native fungi and to test their efficacy to remediate soils from contaminants they are competent for [54], which is intrinsically related to the ambient conditions that these fungi were isolated from. A summary comparing the IC_50_ of Cd and Pb in this study with values reported in the previous literature is provided in Appendix A.

These findings mark one of the first reports of filamentous fungi isolated from soils impacted by mining in Concepcion del Oro, Zacatecas, Mexico, an area that has not been extensively studied in terms of microbial ecology. Additionally, it is significant that the elevated IC_50_ values observed for *Penicillium simplicissimum* (Pb) and *Paecilomyces lilacinus* (Cd and Pb) surpass those reported in previous studies of the same fungi, indicating exceptional tolerance that could reflect long-term adaptation to extreme metal concentrations.

The identification of dual metal tolerance in *P. lilacinus* further underscores its potential as a suitable option for bioremediation efforts in sites affected by multiple metal contaminants. While these experiments were performed under controlled in vitro conditions, the results provide a strong basis for future investigations in more applicable environmental contexts involving highly tolerant organisms isolated from the contaminated areas.

## 5. Conclusions

This study identified six filamentous fungal isolates from soils highly contaminated with mine tailings, which exhibited varying degrees of tolerance to cadmium and lead. Notably, one native strain, *Paecilomyces lilacinus*, demonstrated high tolerance to both metals, suggesting possible physiological or evolutionary adaptations to prolonged heavy metal exposure. Its performance makes it a promising candidate for future remediation trials under controlled conditions and in soil systems. These findings lay the groundwork for the selection of locally adapted fungi for biotechnological applications. Future studies should focus on evaluating their long-term functionality and efficacy through soil experiments and metal removal assays.

## Figures and Tables

**Figure 1 biology-14-00688-f001:**
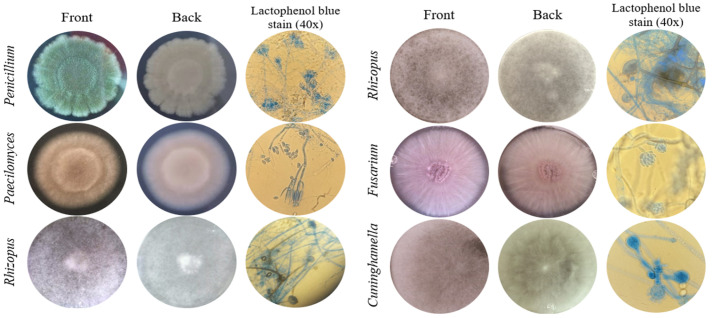
Morphological identification of fungi tolerant to Cd or Pb. Macroscopic and microscopic characteristics of the colonies.

**Figure 2 biology-14-00688-f002:**
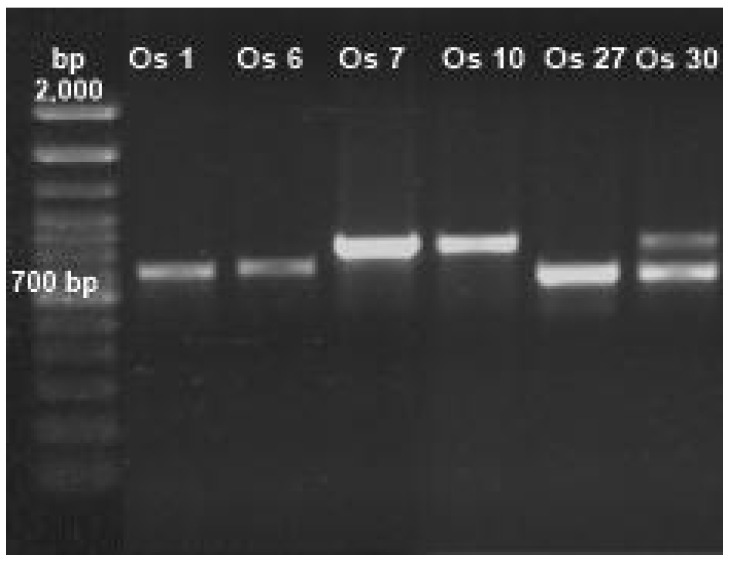
Amplification of internal transcribed spacers (ITSs) 4 and 5 of isolated fungi.

**Figure 3 biology-14-00688-f003:**
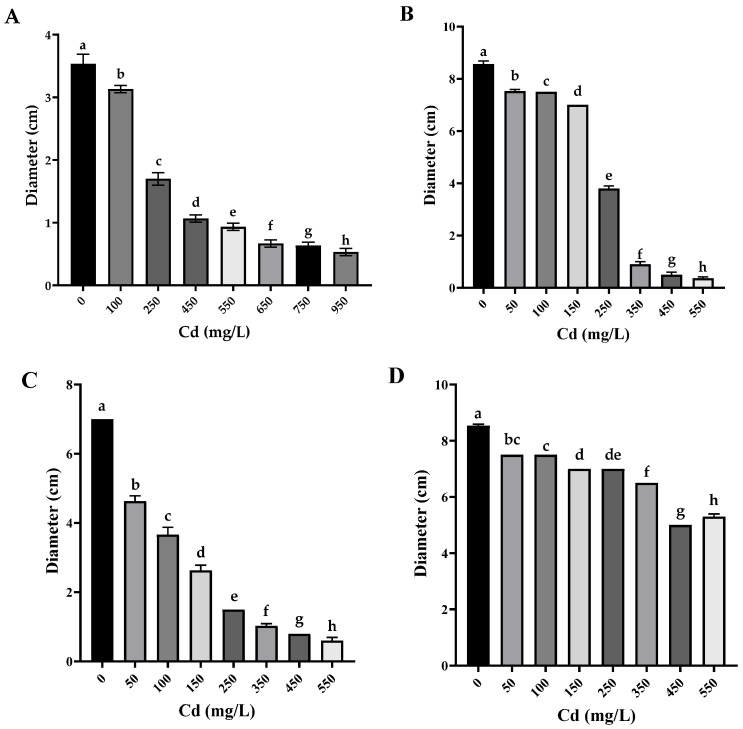
Fungal colony diameter in different concentrations of Cd (*n* = 3). (**A**) *Paecilomyces lilacinus*, (**B**) *Fusarium oxysporum*, (**C**) *Rhizopus microsporus*, and (**D**) *Cuninghamella* sp.; different letters indicate statistically significant differences between means of fungal growth obtained at different cadmium concentrations, including the control (without Cd), according to Tukey’s HSD test (*p* < 0.05).

**Figure 4 biology-14-00688-f004:**
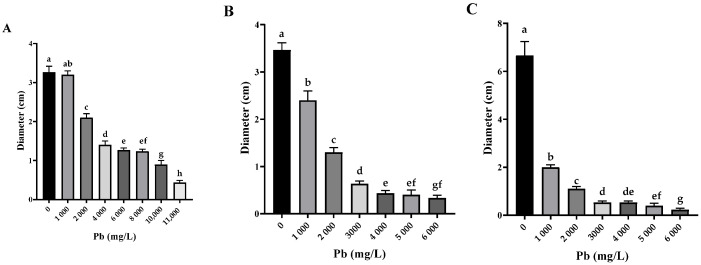
Fungal colony diameters in different concentrations of Pb (*n* = 3). (**A**) Os1 *Penicillium simplicissimum.* (**B**) Os6 *Paecilomyces lilacinus.* (**C**) Os7 *Rhizopus microsporus*; different letters indicate statistically significant differences between the means of fungal growth obtained at different lead concentrations, including the control (without Pb), according to Tukey’s HSD test (*p* < 0.05).

**Figure 5 biology-14-00688-f005:**
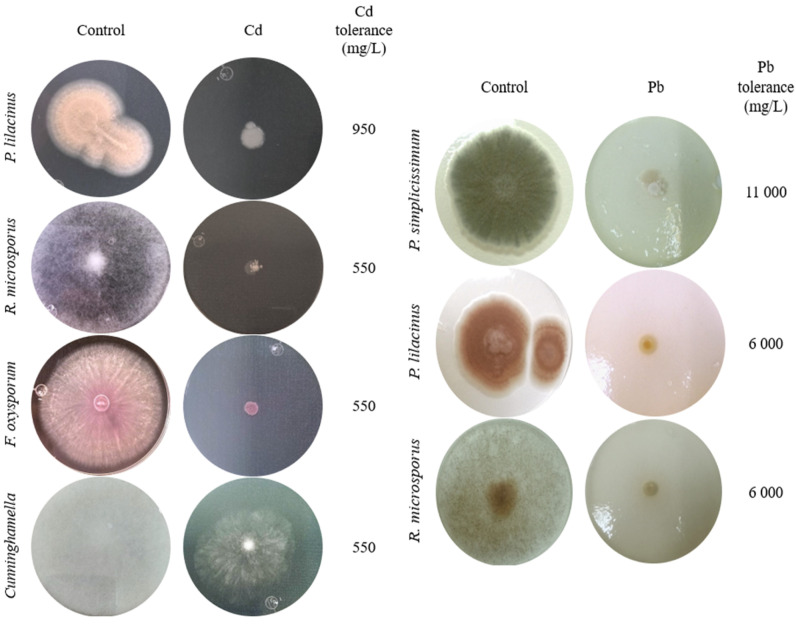
Tolerance and morphological changes in fungal colonies in PDA with and without Cd or Pb.

**Figure 6 biology-14-00688-f006:**
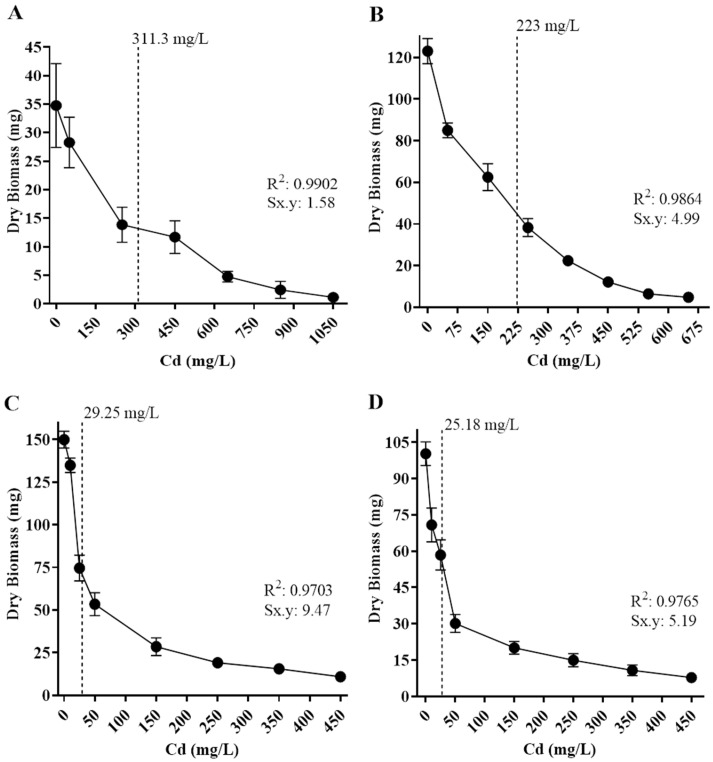
Half-maximal inhibitory concentration (IC_50_) of isolated fungi in different concentrations of Cd. The dotted line corresponds to IC_50_ values. (**A**) *Paecilomyces lilacinus*, 311.3 mg/L; (**B**) *Fusarium oxysporum*, 223 mg/L; (**C**) *Rhizopus microsporus*, 29.25 mg/L; and (**D**) *Cuninghamella* sp, 25.18 mg/L. R^2^ = determination coefficient. Sx.y = standard deviation Cd*dry biomass.

**Figure 7 biology-14-00688-f007:**
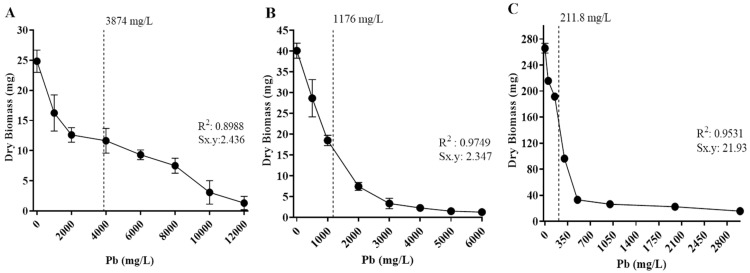
Half-maximal inhibitory concentration (IC_50_) of isolated fungi in different concentrations of Pb. The dotted line corresponds to IC_50_ values. (**A**) *Penicillium simplicissimum*, 3874 mg/L; (**B**) *Paecilomyces lilacinus*, 1 176 mg/L; and (**C**) *Rhizopus microsporus*, 211.8 mg/L. Sx.y: standard deviation dry biomass*Pb.

**Table 1 biology-14-00688-t001:** Morphological and molecular identification of fungi tolerant to Cd or Pb obtained from soils contaminated with mining tailings.

Isolate ID	Morphological Identification	Size (bp)	Molecular Identification	Coincidence (%)	Access
**Os 1** **Os 6** **Os 7**	*Penicillium* sp.	552	*P. simplicissimum*	99.4	MW485753.1
	*Paecilomyces* sp.	641	*P. lilacinus*	99.8	MT453285.1
	*Rhizopus* sp.	664	*R. microsporus*	100	MH473977.1
**Os 10** **Os 27**	*Rhizopus* sp.	666	*R. microsporus*	100	MH473977.1
	*Fusarium* sp.	513	*F. oxysporum*	99.6	KX655587.1
**Os 30**	*Cuninghamella* sp.	715	*Cuninghamella* sp.	87.5	OR096349.1

NCBI (National Center for Biotechnology Information): BLAST: Basic Local Alignment Search Tool 1.4.0 (https://blast.ncbi.nlm.nih.gov/Blast.cgi, Accessed on 22 November 2023); bp: base pairs.

## Data Availability

The original contributions presented in this study are included in the article/Appendix A. Further inquiries can be directed to the corresponding authors.

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
