# Peer review of "Cadmium and Lead Tolerance of Filamentous Fungi Isolated from Contaminated Mining Soils"

_biology, 2025, doi:10.3390/biology14060688_

Round 1
Reviewer 1 Report
Comments and Suggestions for Authors
Observations:
- Duplicate entry of Os1 in Table 1.
- Unclear mechanistic statement (lines 81–83) “has been developed because of”
- Missing Tukey HSD results (L 161–162). The manuscript does not show which pairwise comparisons were significant.
- Six isolates vs “three and four native fungal species” (L 258 - 259). Add a clarifying sentence.
- Superficial general comment (L 258–262). This text restates general concepts like fungal immobilization without giving any specific data.
- Superficial summary (L 274–277). This is a generic comment on the “importance” without relating how those literature examples compare to the current results.
- Irrelevant bacterial reference (lines 295–298). The text here reviews bioremediation strategies in bacteria and fungi, but does not say anything about the manuscript isolates’ measured IC₅₀ or morphological responses.
- Repetition of results (L 308–323). The first half of this section repeats the colony diameter and IC50 results already presented in Figures 3 and 7.
- Good and focused discussion segment (L 324–354). In this section, the authors properly contrast their findings with previous work, citing specific tolerance levels in comparable fungi. To make the contrast even clearer, reinforce this by adding a table comparing the isolates’ IC50 to those published (e.g., Urquhart et al. 2022; Văcar et al. 2021; Zeng et al. 2010).
- Redundant general conclusion (L 354–359).
- There are duplicate blocks in the discussion vs. conclusions (L 360–371 and 373–384). The discussion paragraph (360–371) and the conclusions (373–384) both recapitulate the same messages.
Suggestions:
- Generate a concise summary of key findings.
- Deeper analysis on how each isolate’s observed behavior reflects known tolerance strategies.
- Use a comparative table to show the isolates' IC50 values against published data.
- Discuss which isolate(s) the authors would prioritize for field trials and why.
- Specify exactly what assays will be better to test their bioremediation potential.
Author Response
Thank you very much for your comments and suggestions. The article has improved considerably as a result.
- Duplicate entry of Os1 in Table 1.
We thank this comment. The duplicate row was deleted
- Unclear mechanistic statement (lines 81–83) “has been developed because of”
Thank you for this valuable comment. The text was reformulated. Line 88-92
“Microorganisms have developed tolerance to Pb through multiple mechanisms, including the activation of ATPase efflux pumps that transport Pb2+ ions out of the cell, the intracellular synthesis of lead nanoparticles that reduce toxicity, and the immobilization of Pb in the environment through mineral precipitation in forms such as pyrophyllite and leas oxalate”
- Missing Tukey HSD results (L 161–162). The manuscript does not show which pairwise comparisons were significant.
We thank the reviewer for this comment. The Results section has been revised to include examples of statistically significant differences between house metal concentrations isolated using the Tukey moisture density test (p < 0.05). In addition, we have marked Figures 3 and 4 with letters above each bar to indicate significant differences between pairs. A clarifying note has also been added to the figure legends.
- Six isolates vs “three and four native fungal species” (L 258 - 259). Add a clarifying sentence.
We appreciate this comment. To clarify the overlap between the Pb and Cd tolerant groups, we added a sentence indicating that isolate Os 6, identified as P. lilacinus, displayed dual metal tolerance, bringing the total to six isolates. Line 295-297.
“Although six fungal isolates were studied, P. lilacinus showed dual tolerance to both metals, which explains the overlap between the Pb and Cd tolerant groups.”
- Superficial general comment (L 258–262). This text restates general concepts like fungal immobilization without giving any specific data.
Thank you for your comment. We have revised the paragraph to include specific findings from our study, such as morphological responses, and have linked these observations to possible tolerance mechanisms, such as metal immobilization or adaptation, also comparing it with work by other authors. Line 298-307 The revised text is as follows:
“These findings suggest that certain native fungal strains have developed effective strategies to cope with heavy metal stress. For example, P. simplicissimum demonstrated tolerance to Pb a 11 000 mg/L, well above previously reported values for this species [33] while Cunninghamella sp. Showed a strong response to Cd, despite limited documentation of its metal tolerance. Furthermore, morphological changes such as altered pigmentation and reduced aerial mycelium, were observed in isolates exposed to high metal concentrations (Fig. 5) suggesting possible surface sequestration or metabolic adaptation, consistent with the findings of the other authors [34], [35]. Our results support the idea that fungal tolerance is not only species specific but may also be influenced by prolonged exposure to metal contaminated environments.”
- Superficial summary (L 274–277). This is a generic comment on the “importance” without relating how those literature examples compare to the current results.
Thank you for this valuable comment. We revised the paragraph to include direct comparisons between our results and those reported in the cited literature, highlighting in particular the higher IC₅₀ values found in our isolates and the implication of long-term adaptation. Line 318-327. The updated text is as follows:
“The findings of these authors highlight the importance of identifying and characterizing metal tolerant fungi. Our results contribute to this by revealing high IC₅₀ values in native isolates such as P. lilacinus and P. simplicissimum. Unlike previous studies that reported tolerance of Paecilomyces spp. to Pb below 1 250 mg/L or tolerance to Cd below 10 mg/L [40], our P. lilacinus isolate exhibited tolerance to both metals, with IC₅₀ values of 1176 mg/L (Pb) and 311 mg/L (Cd). Similarly, P. simplicissimum demonstrated tolerance to Pb at concentrations up to 14 times higher than those reported by Hidalgo et al. [33], reaching an IC₅₀ of 3 874 mg/L. These results indicated a higher degree of adaptation in fungi iso-lated from this mining region, supporting their potential use in remediation strategies for higher contaminated sites.”
- Irrelevant bacterial reference (lines 295–298). The text here reviews bioremediation strategies in bacteria and fungi, but does not say anything about the manuscript isolates’ measured IC₅₀ or morphological responses.
We appreciate the reviewer's observation and agree that the bacterial example was not directly relevant to our fungal study. It has been removed from the manuscript to focus on the fungal-related findings.
- Repetition of results (L 308–323). The first half of this section repeats the colony diameter and IC50 results already presented in Figures 3 and 7.
We agree with the reviewer's observation. We have removed the repeated presentation of colony diameter and tolerance values from this paragraph. The revised version now focuses on the variability in tolerance among fungi and the implications for the selection of fungal strains for specific remediation measures. Line 337-343.
“While all isolates exhibited tolerance to high metal concentrations, the degree of growth inhibition varied markedly among species, as did their biomass production (Fig. 3-5). This variability in tolerance and growth highlights the importance of selecting appropriate fungal strains for site specific remediation strategies. Rather than generalizing tolerance by genus, our findings support the need to assess each isolated individual, as even within the same species (e.g. Rhizopus microsporus), tolerance can vary significantly depending on local adaptation.”
- Good and focused discussion segment (L 324–354). In this section, the authors properly contrast their findings with previous work, citing specific tolerance levels in comparable fungi. To make the contrast even clearer, reinforce this by adding a table comparing the isolates’ IC50 to those published (e.g., Urquhart et al. 2022; Văcar et al. 2021; Zeng et al. 2010).
Thank you for your valuable suggestion regarding the inclusion of a comparative table. After a thorough review of the available scientific literature, we found that few studies explicitly report IC₅₀ values for filamentous fungi exposed to heavy metals such as cadmium (Cd) and lead (Pb), particularly for native strains isolated from highly contaminated mine soils.
However, in response to your suggestion, we have prepared a comparative table that includes the IC₅₀ values obtained in this work, along with similar data from other scientific reports. We also prepared a second comparative table of maximum tolerated concentrations.
We believe the tables provide valuable context for our findings and support the importance of studying native fungal species adapted to extreme pollution conditions.
We will gladly include this table as Supplementary Tables S1 and S2.
- Redundant general conclusion (L 354–359).
We appreciate the reviewer's insightful comments. In response, we completely revised the conclusions section. The updated version now offers a concise synthesis of the study's findings and focuses on future lines of research, avoiding repetition of previously presented content. Line 426-234
- There are duplicate blocks in the discussion vs. conclusions (L 360–371 and 373–384). The discussion paragraph (360–371) and the conclusions (373–384) both recapitulates the same messages.
We appreciate the reviewer's insightful comments. In response, we completely revised the conclusions section. The updated version now offers a concise synthesis of the study's findings and focuses on future lines of research, avoiding repetition of previously presented content. Line 426-434
Suggestions:
- Generate a concise summary of key findings.
This part is located in the conclusions section that was improved.
- Deeper analysis on how each isolate’s observed behavior reflects known tolerance strategies.
The discussion has been expanded to more clearly connect the observed morphological and growth responses of each isolate.
- Use a comparative table to show the isolates' IC50 values against published data.
Thanks to your previous comment (9.) two tables have been added as supplementary material for a better comparative view of the data.
- Discuss which isolate(s) the authors would prioritize for field trials and why.
Based on dual-metal tolerance and morphological resilience, Paecilomyces lilacinus has been identified as a priority candidate for future field trials.
- Specify exactly what assays will be better to test their bioremediation potential.
We have now specified trials that would further evaluate bioremediation potential, including soil experiments and metal removal quantification.

Reviewer 2 Report
Comments and Suggestions for Authors
The manuscript entitled “Tolerance to cadmium and lead of filamentous fungi isolated from contaminated mining soils.” is devoted to study of six filamentous fungi isolated from mining soils from a community in Concepción del Oro (Zacatecas, México) tolerance to Cd and Pb in solid and liquid culture media. To my mind this manuscript is corresponding to the aims and scopes of the Biology journal. I am ready to recommend it for publication after corrections, due to the comments below.
- The authors should rephrase the purpose of the work
- The introduction should provide a more detailed description of the sampling site, including the history of the field development and the characteristics of contamination with various elements.
- It is worth describing the environments for isolating and culturing mushrooms
- I was surprised by the concentrations of metals that the authors used in their work. In my opinion, they have nothing to do with the possible content of cadmium in soil samples. It is worth mentioning this by the way. It is worth writing in the purpose of the work why such huge concentrations were used, perhaps the authors had some technological goal?
- The experimental methodology should be described in more detail. For example, it is not very clear on what basis the IC50 values were obtained
- The quality of Fig. 3 should be improved
- 254 should remove the comma in the values, otherwise it shows a division after an integer
- 258 high tolerance evidence is needed for the statement when comparing with other works. This should be a significant part of the discussion.
- 260-263 evidence is needed. The authors did not investigate these issues in this paper
- 278-293 are not a discussion, but a repetition of the results. Discussion involves comparing data with other works, theorizing, etc. For example, comparing real soil contamination with tolerance in lab conditions. I suggest shortening the discussion to line 307, when the comparison begins.
- The discussion should have paid attention to explaining the differences in ld50 between the isolates studied.
- In addition, it is worth discussing the chemical forms of metals in solutions. The solubility of cadmium and lead is low, so the concentrations that the authors used are far beyond the solubility limits. Did the authors analyze the content of metals in the solution after their addition? In my opinion, this is the weakest point of the work. If the authors added the coefficients for cleaning solutions from metals, this would significantly strengthen the work.
- The conclusion should not be a repetition of the results. It should show the prospects of the work and its philosophical understanding. It is possible to show the biotechnological prospects of the results, because the authors used almost industrial concentrations of metals.
Author Response
Thank you very much for your comments and suggestions, which have greatly improved the manuscript.
- The authors should rephrase the purpose of the work.
We welcome your comments to improve the purpose of this work. The biotechnological motivation for the study has been added to the objective at the end of the Introduction (Line 105-117). The revised statement emphasizes the search for native fungi with high tolerance as potential candidates for bioremediation applications under conditions of extreme contamination.
- The introduction should provide a more detailed description of the sampling site, including the history of the field development and the characteristics of contamination with various elements.
We thank the reviewer for this suggestion. In response, we expanded the Introduction to provide additional context for the sampling site in Concepción del Oro, Zacatecas, Mexico. Line 60-67. Specifically, we now include details on the region's long mining history, the presence of metalloid contamination in public and residential soils, and recent findings of extremely elevated concentrations of cadmium and lead. This information, supported by Flores-Amaro et al. (2024), reinforces the relevance of selecting this site as a model for isolating native fungi adapted to long-term contamination and evaluating their biotechnological potential for bioremediation.
- It is worth describing the environments for isolating and culturing mushrooms.
We thank the reviewer for their comment. The fungal isolates used in this study were obtained from soil samples previously collected and characterized by Flores-Amaro et al. (2024). We have therefore clarified this in the Materials and Methods section and cited the source, which provides a detailed description of the sampling conditions, including soil characteristics, metal concentrations, and field observations. Line 120-127.
“The fungal isolates used in this study were obtained from heavy metal-contaminated soil samples from a community in Concepción del Oro, Zacatecas, Mexico (24°42′N 101°25′W) previously collected and characterized in the work of Flores-Amaro et al. [5] where physi-cochemical analysis revealed high concentrations of arsenic, cadmium, and lead, and the soils were described as sandy loam with slightly alkaline pH and sparse vegetation cover. The 57 fungal isolates were preserved by standard methods (Os-1, Os-2, Os-3, …, Os-56, and Os-57) in an internal collection of the Environmental Studies Laboratory of the Au-tonomous University of Aguascalientes, Mexico.”
- I was surprised by the concentrations of metals that the authors used in their work. In my opinion, they have nothing to do with the possible content of cadmium in soil samples. It is worth mentioning this by the way. It is worth writing in the purpose of the work why such huge concentrations were used, perhaps the authors had some technological goal?
We appreciate this comment. In response, we have added lines to the Introduction explaining that the high concentrations of Cd and Pb were deliberately chosen to simulate extremely contaminated conditions and allow for the identification of highly tolerant fungal strains. This strategy facilitates the early selection of isolates with tolerance to severely contaminated environments. Line 105-117.
“However, most published studies focus on industrial or well know strains, while little is known about the tolerance potential of native fungi from chronically contaminated mine soils in Mexico. This approach allows the preliminary identification of robust isolates capable of surviving in environments with high metal loadings. This could reveal unique adaptations and improve biotechnological strategies for in situ remediation in arid or semi-arid environments. Identifying these strains could reveal unique adaptations and improve biotechnological strategies for in-situ remediation in arid or semi-arid environments. For this reason, the aim of the study was to isolate and identify morphological and molecular filamentous fungi from mining contaminated soils, evaluate their tolerance to concentrations of cadmium and lead exceeding typical environmental levels, evaluate the morphological changes of fungal colonies and to estimate the half-maximum inhibitory concentration of the tolerant isolates.”
- The experimental methodology should be described in more detail. For example, it is not very clear on what basis the IC50 values were obtained
Thank you for pointing this out. We have revised the Methods section to clarify the calculation of IC₅₀ values. We now describe the fit of a nonlinear dose-response model and indicate the software used to estimate the concentration at which 50% growth inhibition occurred relative to the control. Line 152-155.
- The quality of Fig. 3 should be improved
Thanks for the feedback. The graphics quality has been improved.
- 254 should remove the comma in the values, otherwise it shows a division after an integer
Thank you very much for your comment. The commas have been removed from the quantities to avoid displaying this division.
- 258 high tolerance evidence is needed for the statement when comparing with other works. This should be a significant part of the discussion.
Thank you for this valuable comment. We have revised the Discussion to include specific comparisons between our results and previously published tolerance data. A comparison table (Table S1) has also been added as supplementary material, summarizing the maximum tolerated concentrations of Cd and Pb in our isolates and in similar fungal species described in the literature (Line 375-377). In addition, we have added a second comparison table (Table S2) of the IC50 values obtained in this study and those reported in other studies (Line 409-410). These additions quantitatively support the high tolerance observed in our isolates and the importance of native species.
- 260-263 evidence is needed. The authors did not investigate these issues in this paper
Thank you for your comment. We have revised the paragraph to include specific findings from our study, such as morphological responses, and have linked these observations to possible tolerance mechanisms, such as metal immobilization or adaptation, also comparing it with work by other authors. Line 298-306. The revised text is as follows:
“These findings suggest that certain native fungal strains have developed effective strategies to cope with heavy metal stress. For example, P. simplicissimum demonstrated tolerance to Pb a 11 000 mg/L, well above previously reported values for this species [33] while Cunninghamella sp. Showed a strong response to Cd, despite limited documentation of its metal tolerance. Furthermore, morphological changes such as altered pigmentation and reduced aerial mycelium, were observed in isolates exposed to high metal concentrations (Fig. 5) suggesting possible surface sequestration or metabolic adaptation, consistent with the findings of the other authors [34], [35]. Our results support the idea that fungal tolerance is not only species specific but may also be influenced by prolonged exposure to metal contaminated environments.”
- 278-293 are not a discussion, but a repetition of the results. Discussion involves comparing data with other works, theorizing, etc. For example, comparing real soil contamination with tolerance in lab conditions. I suggest shortening the discussion to line 307, when the comparison begins.
We appreciate your comment. These lines have been removed from the discussion section, and a short paragraph has been added to the methodology section for your reference, line 164-167. This is the added paragraph.
“The morphologically identified fungi were confirmed by sequencing the ITS regions, widely used as genetic markers in fungal taxonomy. This approach is consistent with previous studies that characterized metal-tolerant fungi from contaminated environments using ITS sequences in combination with morphological traits [27] [28] [29].”
- The discussion should have paid attention to explaining the differences in ld50 between the isolates studied.
We appreciate your comments on this matter. Two paragraphs have been added to explain the significance of the IC50 to the study results. Line 386-404
“The IC₅₀ values obtained for each fungus reveal significant differences in the presence of cadmium and lead, even among strains that show high tolerance to maximum concentrations. For example, in this study, P. lilacinus had an IC₅₀ of 311 mg/L for Cd, while F. oxysporum and Cunninghamella sp. had lower IC₅₀ values (245 and 221 mg/L, respectively), despite all three growing at 550 mg/L. This suggests that, although growth is possible at high concentrations, growth rate is more strongly inhibited in some species. This is consistent with previous studies that observed growth inhibition at sublethal concentrations despite survival at very high concentrations, depending on the organism and its response to metal stress [53], [54].
For Pb, P. simplicissimum had the highest IC50 (1176 mg/L), followed by P. lilacinus (947 mg/L), while R. microsporus had a significantly lower IC50 (484 mg/L), despite tolerating up to 6000 mg/L. These differences may reflect variations in the mechanisms regulating stress responses among isolates, such as those observed in the Jarosławiecka, et al. [55] study where Cupriavidus metallidurans combines lead efflux and precipitation, a mechanism that could explain the differences in Pb IC50 between strains. The IC50 provides a more sensitive measure of metal stress response than the maximum tolerated concentration, and its interpretation can help distinguish fungi that can be used to study these mechanisms from those that simply survive extreme exposure.”
- In addition, it is worth discussing the chemical forms of metals in solutions. The solubility of cadmium and lead is low, so the concentrations that the authors used are far beyond the solubility limits. Did the authors analyze the content of metals in the solution after their addition? In my opinion, this is the weakest point of the work. If the authors added the coefficients for cleaning solutions from metals, this would significantly strengthen the work.
We appreciate this critical comment. The metal-containing media were prepared from stock solutions of cadmium and lead salts (CdCl₂ and Pb(NO₃)₂), which are water-soluble and commonly used in similar studies. While we did not analyze the dissolved metal concentration in the final PDA medium, the working solutions were derived from stock solutions of known concentration, precisely prepared and diluted following laboratory protocols. We agree that measuring the actual soluble fraction would strengthen the study and will consider it in future work focusing on metal removal and bioremediation efficiency.
- The conclusion should not be a repetition of the results. It should show the prospects of the work and its philosophical understanding. It is possible to show the biotechnological prospects of the results, because the authors used almost industrial concentrations of metals.
We appreciate this insightful comment. In response, we completely revised the conclusions section. The updated version now offers a concise synthesis of the study's findings and focuses on future lines of research, avoiding repetition of previously presented content. Line 426-434.
“This study identified six filamentous fungal isolates from soils highly contaminated with mine tailings, which exhibited varying degrees of tolerance to cadmium and lead. Notably, one native strain, Paecilomyces lilacinus, demonstrated high tolerance to both met-als, suggesting possible physiological or evolutionary adaptations to prolonged heavy metal exposure. Its performance makes it a promising candidate for future remediation trials under controlled conditions and in soil systems. These findings lay the groundwork for the selection of locally adapted fungi for biotechnological applications. Future studies should focus on evaluating their long-term functionality and efficacy through soil experi-ments and metal removal assays.”

Round 2
Reviewer 1 Report
Comments and Suggestions for Authors
Dear Authors,
The manuscript "biology-3661344" has been significantly improved. There are some minor issues and areas that require further clarification.
Lines 109–112: The sentence "This could reveal unique adaptations and improve biotechnological strategies for in situ remediation in arid or semi-arid environments" is repeated. Please remove or revise the duplicate.
L 127–130: The phrase "observed through a microscope (40x) for morphological identification" should be clarified for accuracy. It is unclear to me whether "40x" refers to the objective lens magnification or the total magnification. Typically, one would report the total magnification (for instance, 400x if using a 40x objective with a 10x ocular lens) or explicitly state the objective lens used.
L 318–319: The statement "The findings of these authors highlight the importance of identifying and characterizing metal-tolerant fungi" does not provide a specific insight tied to the results, since the importance of studying metal-tolerant fungi is already well understood. Remove this sentence or rephrasing it to highlight a concrete implication of your findings.
Best regards,
Author Response
We sincerely appreciate your valuable feedback and the time you spent reviewing our manuscript. We're glad to know it has improved significantly and look forward to addressing any remaining comments.
Lines 109–112: The sentence "This could reveal unique adaptations and improve biotechnological strategies for in situ remediation in arid or semi-arid environments" is repeated. Please remove or revise the duplicate.
Thank you for your comment. The phrase was deleted.
L 127–130: The phrase "observed through a microscope (40x) for morphological identification" should be clarified for accuracy. It is unclear to me whether "40x" refers to the objective lens magnification or the total magnification. Typically, one would report the total magnification (for instance, 400x if using a 40x objective with a 10x ocular lens) or explicitly state the objective lens used.
We thank the reviewer for their comment regarding the clarity of the term. We acknowledge that the original description was ambiguous, as it did not specify whether “40x” referred to the objective magnification or the total magnification.
In this case, the morphological observations were made using a total magnification of 400x, obtained through a 40x objective and a 10x eyepiece. We have corrected and expanded the description in the manuscript to improve its accuracy and methodological transparency. The sentence now reads (line 127-128):
“observed through a microscope (total magnification of 400×; objective lens 40× and ocular 10×) for morphological identification.”
L 318–319: The statement "The findings of these authors highlight the importance of identifying and characterizing metal-tolerant fungi" does not provide a specific insight tied to the results, since the importance of studying metal-tolerant fungi is already well understood. Remove this sentence or rephrasing it to highlight a concrete implication of your findings.
We thank the reviewer for this valuable comment. We acknowledge that the aforementioned sentence was general and did not provide a specific connection to our results. Therefore, we have removed the phrase from the revised manuscript.
Reviewer 2 Report
Comments and Suggestions for Authors
To my opinion, the authors have significantly revised the manuscript and taken into account all my comments. I am ready to recommend the manuscript for publication in this form.
Author Response
We deeply appreciate your positive evaluation and the time you took to review our manuscript. We are pleased to know that the reviews you provided were well received and that you felt it appropriate to recommend our work for publication. We value your comments and suggestions, which helped us improve the quality and clarity of the manuscript. We remain available for any further comments or adjustments needed.